

# 3D Ising CFT and exact diagonalization on icosahedron: The power of conformal perturbation theory

**Bing-Xin Lao[1,2]⋆ and Slava Rychkov[2]†**

**1** École Normale Supérieure - PSL, 45 rue d'Ulm, F-75230 Paris cedex 05, France
**2** Institut des Hautes Études Scientifiques, 91440 Bures-sur-Yvette, France

⋆ bingxin.lao@ens.psl.eu ,   † slava@ihes.fr

## Abstract

We consider the transverse field Ising model in $(2+1)$D, putting 12 spins at the vertices of the regular icosahedron. The model is tiny by the exact diagonalization standards, and breaks rotation invariance. Yet we show that it allows a meaningful comparison to the 3D Ising CFT on $\mathbb{R} \times S^2$, by including effective perturbations of the CFT Hamiltonian with a handful of local operators. This extreme example shows the power of conformal perturbation theory in understanding finite $N$ effects in models on regularized $S^2$. Its ideal arena of application should be the recently proposed models of fuzzy sphere regularization.



# 1  Introduction

A $d$-dimensional conformal field theory (CFT) on $\mathbb{R} \times S^{d-1}$ is Weyl-equivalent to the same theory on $\mathbb{R}^d$.[1] For $d = 2$, this leads to a very efficient method to compute CFT data by diagonalizing critical spin chain Hamiltonians or critical transfer matrices on finely discretized $S^1$ [2–4].[2] For $d = 3$ this has been very little studied until recently, because $S^2$ is harder to discretize than $S^1$, and rotation invariance is hard to recover [6–9]. Recently, Ref. [10] considered a Hamiltonian on $S^2$ realizing a quantum phase transition in the $(2 + 1)$D Ising universality class, which preserves *exact* rotation invariance. Already for small systems $N \leq 24$, where $N$ is the number of electrons on the sphere, results were obtained for CFT operator dimensions [10], operator product expansion (OPE) coefficients [11], and the four-point functions [12] which compare well with the conformal bootstrap [13–16], up to small deviations associated to finite $N$ effects.

Why does the model of [10] work so well? One possibility is that the model is somehow very special (beyond the fact that it realizes exact rotation invariance). Another possibility is that it is the 3D Ising CFT which is special, and any model approximating it will do so rather well. The latter is favored by the sparsity of the low-lying spectrum of the 3D Ising CFT. In the $\mathbb{Z}_2$ even scalar sector, after the relevant $\epsilon$ of dimension $\Delta_\epsilon \approx 1.41$ and the leading irrelevant $\epsilon'$ of dimension $\Delta_{\epsilon'} \approx 3.83$ the next irrelevant operator has a rather high dimension $\Delta_{\epsilon''} \approx 6.89$ [17]. Given that [10] performs a double tuning of the model's parameters, a possible scenario is that the operators $\epsilon$ and $\epsilon'$ are both tuned away, while the corrections associated with $\epsilon''$ are expected to scale as $1/N^\alpha$ with a high power $\alpha = \frac{1}{2}(\Delta_{\epsilon''} - 3)$, and may be small even at moderate $N$.

These considerations can be more constructively formulated as the following

**Conjecture 1.** *Finite $N$ effects in the spectrum on $S^2$ of the model of [10] can be understood by an effective theory, perturbing the CFT Hamiltonian by integrals of local CFT operators times small couplings.*

In this paper we will show that a similar conjecture works even for a much simpler model *without* rotation invariance - the transverse field Ising model (TFIM) on the icosahedron. Preliminary results of this work were reported in [18]. The Hamiltonian of the model is given by

$$H_{\text{TFIM}} = -J \sum_{\langle ij \rangle} \sigma_i^z \sigma_j^z - h \sum_i \sigma_i^x \,, \tag{1}$$

---

[1]This means that correlation functions $\mathbb{R} \times S^{d-1}$ can be obtained from correlation functions on $\mathbb{R}^d$ via a Weyl transformation. In this work we will focus on the 3D Ising CFT. In $d = 3$, as in any odd $d$, the equivalence follows easily because there is no Weyl anomaly. For even $d$ the equivalence was proved for unitary CFTs in $d \leq 10$ dimensions [1].

[2]See [5], Chapter 3, for a review and many more references.

where $i$ runs over 12 icosahedron vertices, and $\langle ij \rangle$ over 30 icosahedron edges. The icosahedron is chosen because its spatial symmetry group is the largest irreducible discrete subgroup of $O(3)$. Thus the Hamiltonian (1) is "as close as one can get" to rotation invariance via naive discretization on a regular grid.[3] Ref. [10] achieves exact rotation invariance via an alternative smart construction of "fuzzy sphere regularization" which we will not use here. Our main point here will be that, even with a broken rotation invariance and in a very small model with only 12 spins on the sphere, we will still be able to understand deviations of the exact diagonalization spectra from CFT via an effective theory.

Once the above conjecture is verified for the icosahedron model, it should be plausible that a similar procedure will work for the more sophisticated model of [10]. This will be demonstrated in [19].[4] Until now, the finite $N$ effects in the model of [10] were minimized by tuning and by increasing $N$. Applying effective theory on top of this tuning should allow a significant increase in the accuracy of CFT data extraction, as we discuss in the conclusions.

## 2 Exact diagonalization

Hamiltonian (1) commutes with the global $\mathbb{Z}_2$ spin flip generated by $\prod_i \sigma_i^x$. It is also invariant under the spatial icosahedral symmetry $I_h \cong A_5 \times \mathbb{Z}_2^{O(3)}$. We imagine the icosahedron inscribed into the unit sphere centered at the origin. Then $A_5 \subset SO(3)$ while the spatial $\mathbb{Z}_2^{O(3)}$ acts as $x \rightarrow -x$. Finally, the Hamiltonian is real and thus has time reversal symmetry $T$ acting as complex conjugation.

The Hilbert space of model (1) has dimension 4096, and the spectrum is easy to compute via exact diagonalization, see Fig. 1. Let us describe some of its salient features. All eigenstates are either $\mathbb{Z}_2$ odd (blue) or $\mathbb{Z}_2$ even (red) with respect to the global $\mathbb{Z}_2$ spin flip generated by $\prod_i \sigma_i^x$. All eigenvalues have exact degeneracy 1,3,4, or 5, which are the dimensions of irreducible representations of $A_5$, see App. A.

It is known that the TFIM on an infinite *plane*, i.e. on an infinite regular (say, cubic) planar lattice shows a quantum phase transition in the $(2+1)$D Ising universality class at a critical value $h = h_c$.[5] This phase transition separates the phase of the spontaneously broken $\mathbb{Z}_2$ symmetry at $h < h_c$, where the ground state, in infinite volume, is doubly degenerate, and the preserved $\mathbb{Z}_2$ symmetry at $h > h_c$. On a *finite* lattice of size $L \times L$, the energy gaps $\hat{E}_i = E_i - E_0$ approach zero at $h = h_c$ as $\sim L^{-1}$ for $L \rightarrow \infty$. In particular the lightest $\mathbb{Z}_2$ even state gap has a characteristic dip close to $h = h_c$, Fig. 2.

Some of these features are visible in the icosahedron spectrum in Fig. 1. In particular, we see that the lightest $\mathbb{Z}_2$ odd state is almost degenerate with the vacuum for $h \lesssim 3$ and the lightest $\mathbb{Z}_2$ even gap shows a dip around the same position, around $h \sim 3$.

### 2.1 Adjusting the speed of light

We will define $h_c$ in our model as the point where the icosahedron spectrum is the closest to the 3D Ising CFT spectrum. One could have guessed that $h_c \sim 3$ from the dip in the lightest $\mathbb{Z}_2$ even gap, but as we will now see the true $h_c$ is quite a bit higher. This should not be surprising - the dip is very shallow, because we only have 12 points on the sphere.

---

[3]Adding more points on the icosahedron faces [6] does not increase the symmetry of the model.

[4]The word "fuzzy" in the name of the model of [10] may suggest that the model is potentially nonlocal. However, as already explained in [10], only fermions in their model feel the non-commutative structure on the sphere permeated by the magnetic flux, after projection to the first Landau level. On the other hand, the Ising order parameter field is a fermion bilinear and is not sensitive to non-commutativity. Thus the critical point, in the Ising universality class, is expected to be a local CFT.

[5]See e.g. [20], Chapter 5, for an introduction, and [21] for the critical magnetic field determination.

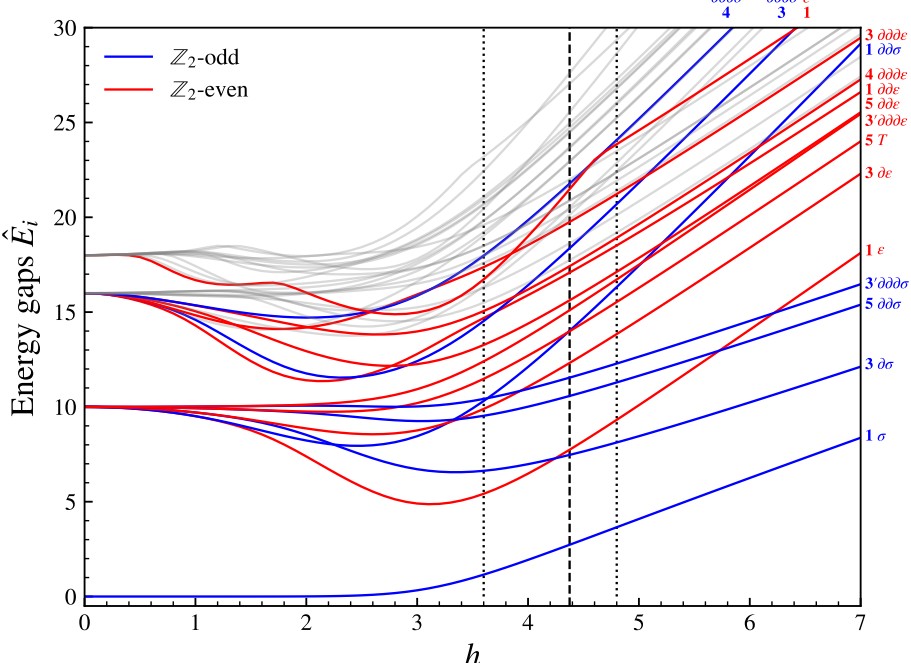

Figure 1: Low-lying spectrum of (1) for $J = 1$ and $h \in [0, 7]$. We plot energy gaps $\hat{E}_i = E_i - E_0$ ($i = 1, 2, \ldots$) where $E_0$ is the ground state energy (not shown). Numbers $\mathbf{1}, \mathbf{3}, \mathbf{3}', \mathbf{4}, \mathbf{5}$ next to the lines indicate the $A_5$ representation. States in color are the states which will be used in the analysis in this paper. Labels $\sigma, \epsilon, \partial\sigma, \partial\epsilon, \ldots$ indicate their identification with CFT states, see below. Color indicates the $\mathbb{Z}_2$ quantum number. Further states are shown in gray to avoid the clutter; they also have well-defined $\mathbb{Z}_2$. All eigenstates also have well-defined $\mathbb{Z}_2^{O(3)}$ (not shown). In the analysis below we will focus on the range of $h$ within the vertical dotted lines. The vertical dashed line is at $h = 4.375$ where $g_\epsilon$ crosses 0 in Fig. 7(b).

The 3D CFT energy levels on $\mathbb{R} \times S^2$ are in one-to-one correspondence with the scaling dimensions of CFT operators: $E_i^{\text{CFT}} = \Delta_i$. This equation has to be translated to our exact diagonalization context as:

$$\alpha \hat{E}_i = \Delta_i + \ldots, \tag{2}$$

with some constant $\alpha$ independent of $i$, which reflects arbitrary normalization of the Hamiltonian. Alternatively, one can think of $\alpha$ as the choice of units of time (or of energy) which is necessary to restore the local 3D isotropy of $\mathbb{R} \times S^2$, inherited from $\mathbb{R}^3$ via Weyl transformation. In condensed matter literature $\alpha^{-1}$ is referred to as the "speed of light" parameter of a quantum critical point.

The terms $\ldots$ in (2) stand for correction terms coming from the perturbations of the CFT Hamiltonian, which will be discussed below. Ignoring these terms for now and taking the ratio of, say, the first $\mathbb{Z}_2$ odd and $\mathbb{Z}_2$ even energy levels, which should be related to 3D Ising CFT operators $\sigma$ and $\epsilon$, the speed of light cancels, and we get

$$\hat{E}_1 / \hat{E}_2 \approx \Delta_\sigma / \Delta_\epsilon \approx 0.367 \qquad (h = h_c), \tag{3}$$

where in the second equation we used the values of $\Delta_\epsilon$ and $\Delta_\sigma$ from the conformal bootstrap, see App. B.

From this equation we find $h_c \approx 4.5$.

Let us test this prediction with more states. Working in a window $h \in [3.6, 4.8]$ we consider the first two levels of degeneracy 1 and 3, in $\mathbb{Z}_2 = \pm$ sectors. These levels should correspond



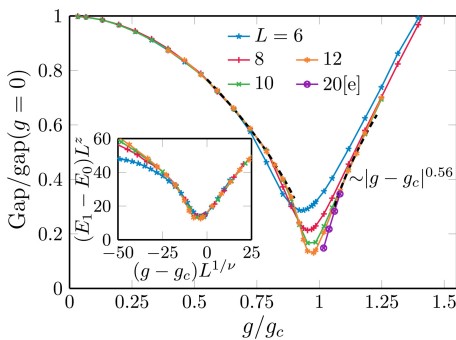

Figure 2: The lightest $\mathbb{Z}_2$ even state gap in the TFIM on the $L \times L$ square lattice [22].[6] This paper denotes $g, g_c$ for our $h, h_c$. On the square lattice $h_c = 3.04438$ [21].

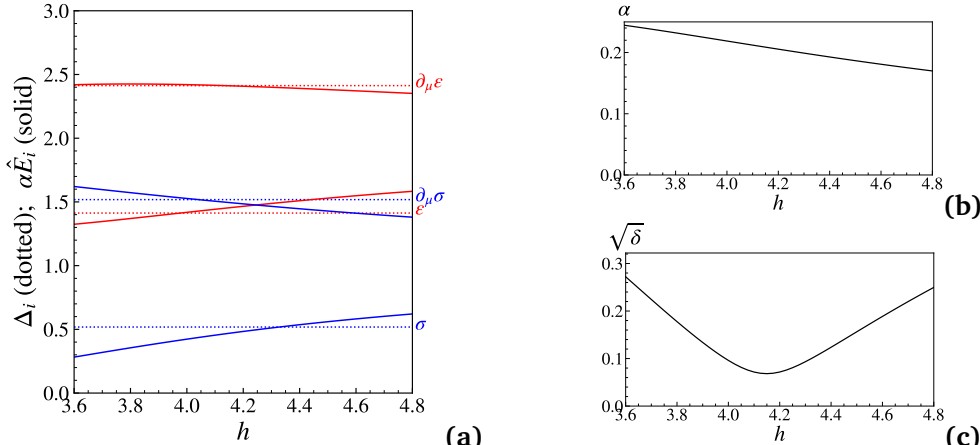

Figure 3: **(a)** Dotted: CFT states $\sigma, \partial_\mu \sigma, \epsilon, \partial_\mu \epsilon$. Solid: $\alpha \hat{E}_i$ for the corresponding exact diagonalization energy gaps, with the best fit $\alpha$. **(b)** Best fit $\alpha$. **(c)** Fit quality.

to operators $\sigma, \partial_\mu \sigma, \epsilon, \partial_\mu \epsilon$, of dimensions $\Delta_\sigma, \Delta_\sigma + 1, \Delta_\epsilon, \Delta_\epsilon + 1$. We perform, for each $h$, a fit for $\alpha$ minimizing the quantity

$$\delta = \sum_i (\alpha \hat{E}_i - \Delta_i)^2, \tag{4}$$

where $i$ runs over these 4 states.

The result is shown in Fig. 3, where horizontal dotted lines show the four exact CFT energy levels, and solid curves show $\alpha \hat{E}_i$ for the best fit value of $h$. We see that the rescaled exact diagonalization gaps are close to the CFT energy levels, with the best agreement at around $h \sim 4.2$, somewhat below the above $h_c$ estimate 4.5. However, the agreement is clearly far from perfect. For example the correct ordering between the energy levels corresponding to $\partial_\mu \sigma$ and $\epsilon$ is not very well reproduced by the numerical data. We will now describe a more sophisticated theoretical effective model, which will lead to a much better agreement.

## 3   Perturbations of the CFT Hamiltonian

We consider the 3D Ising CFT on $\mathbb{R} \times S^2$. Recall that the CFT Hilbert space states on $S^2$ is in one-to-one correspondence with the local CFT operators $\mathcal{O}$, and we will label them as $|\mathcal{O}\rangle$.

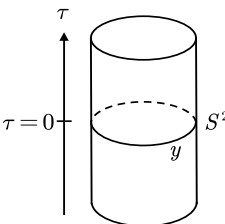

Figure 4: The Hamiltonian perturbation will be constructed by integrating local CFT operators over the $\tau = 0$ slice of the cylinder $\mathbb{R} \times S^2$.

Here $\mathcal{O}$ can be a primary operator, or a derivative of a primary operator (descendant). The CFT Hamiltonian $H_{\text{CFT}}$ is diagonal in this basis, with eigenvalue the scaling dimension:

$$H_{\text{CFT}} |\mathcal{O}\rangle = \Delta_{\mathcal{O}} |\mathcal{O}\rangle . \tag{5}$$

To describe the spectrum of the TFIM on the icosahedron, we will consider an effective Hamiltonian which will be a perturbation of the CFT Hamiltonian:

$$H = H_{\text{CFT}} + \delta H . \tag{6}$$

The CFT Hamiltonian preserves global $\mathbb{Z}_2$, spatial $O(3)$, as well as time reversal. The perturbation $\delta H$ has to preserve symmetries of $H_{\text{TFIM}}$: global $\mathbb{Z}_2$, spatial $I_h$, and time reversal.

We will construct $\delta H$ by integrating local operators of the 3D Ising CFT over the sphere. Let $g(y)$ be a real function on the sphere. Then our $\delta H$ will be a sum of terms having the form:

$$\int_{S^2} g(y) \mathcal{V}(0, y), \tag{7}$$

where $\mathcal{V}(0, y)$ is a local operator located at the $\tau = 0$ slice of the cylinder $\mathbb{R} \times S^2$ and $y$ is a coordinate on $S^2$ (see Fig. 4). Here and below all integrals over $S^2$ are with the standard uniform metric.

Note that $H_{\text{CFT}}$ itself can also be written in this form, as the integral of the stress tensor component $T^{\tau\tau}$.

We will choose $\mathcal{V}$ to be a primary operator, i.e. without derivatives. Derivatives along the sphere can be integrated by parts. Derivatives in the $\tau$ direction give $\delta H$ which have zero diagonal matrix elements. In this paper we will only do first-order perturbation theory so only diagonal matrix elements will be of interest.[7]

To preserve global $\mathbb{Z}_2$, we will consider $\mathbb{Z}_2$-even operators $\mathcal{V}$.

In case of scalar $\mathcal{V}$, further symmetry requirements are as follows. To preserve spatial $I_h$, the coupling function $g(y)$ will have to be $I_h$ invariant. We assume that $\mathcal{V}$ has even intrinsic spatial parity, as is the case for all low-lying primary operators of the 3D Ising CFT. Then (7) preserves time reversal.

The 3D Ising CFT also contains low-lying symmetric traceless primaries of even spin $\ell$, and $\mathcal{O}$ can be one of those. In that case $g(y)\mathcal{V}(0, y)$ in (7) should be understood as $g_{\mu_1 \ldots \mu_\ell}(y)\mathcal{V}_{\mu_1 \ldots \mu_\ell}(0, y)$. Indices $\mu_i$ can point along the sphere and along the $\tau$ direction. Time reversal $\tau \to -\tau$ requires that $g_{\mu_1 \ldots \mu_\ell}(y)$ vanishes whenever an odd number of indices

---

[7]More generally, $\tau$-derivatives correspond to redundant deformation of the Hamiltonian which do not change the energy spectrum. Indeed, performing a unitary transformation $H \to e^{-iV} H e^{iV}$ and expanding to first order in $V$, we find $H \to H + i[H, V] = H + i\partial_\tau V$, i.e. deformation by a total $\tau$ derivative. This redundancy echoes the redundancy of total derivative deformations in the Lagrangian formalism.

equals $\tau$. Furthermore, using tracelessness of $\mathcal{V}$, we may assume without loss of generality that $g_{\mu_1...\mu_\ell}(y)$ is nonvanishing only if all indices $\mu_i$ are along the sphere. To preserve spatial $I_h$, this tensor function has to be $I_h$ covariant.

In the next sections we will see how adding perturbations of the form (7) we will be able to achieve better and better description of the exact diagonalization spectrum of the TFIM on the icosahedron.

## 4   Numerical tests

### 4.1   Adding $\epsilon$ perturbation

On physical grounds, we expect that the most important perturbation is that of $\mathcal{V} = \epsilon$, which is the only relevant $\mathbb{Z}_2$ even scalar primary. We are interested in the effect of this perturbation on the CFT energy levels. We will assume that the coupling function $g(y)$ is small and apply the first-order Hamiltonian perturbation theory:

$$\delta E_\psi = \frac{\langle \psi | \delta H | \psi \rangle}{\langle \psi | \psi \rangle} \,. \tag{8}$$

We will be considering states $|\psi\rangle$ related to the primary operators $|\mathcal{O}\rangle$ and to their descendants. Mapping the cylinder $\mathbb{R} \times S^2$ to $\mathbb{R}^3$, the necessary matrix elements can be related to the three-point (3pt) functions where $\mathcal{O}$ is inserted at 0 and $\infty$ and $\mathcal{V}$ is integrated over the unit sphere. These computations are carried out in App. C. Here we will just describe the general features and present the needed results. The function $g(y)$ will typically enter the answer only through its overall integral over the sphere

$$g_\mathcal{V} := \int_{S^2} g(y) \,, \tag{9}$$

as an overall proportionality factor. Furthermore, the matrix element for $|\psi\rangle = |\mathcal{O}\rangle$ will be proportional to the OPE coefficient $f_{\mathcal{O}\mathcal{V}\mathcal{O}}$, while the matrix elements for $|\psi\rangle$ descendants of $|\mathcal{O}\rangle$ - to the same OPE coefficient times a factor depending on $\Delta_\mathcal{O}$ and $\Delta_\mathcal{V}$, which is fixed by conformal invariance.

When perturbation $\mathcal{V}$ is a primary scalar, and $\mathcal{O}$ is also a scalar, we have (see (C.2), (C.8)):

$$\delta E_\mathcal{O} = g_\mathcal{V} f_{\mathcal{O}\mathcal{V}\mathcal{O}} \,,$$
$$\delta E_{\partial\mathcal{O}} = g_\mathcal{V} f_{\mathcal{O}\mathcal{V}\mathcal{O}} A_{\partial\mathcal{O},\mathcal{V}} \,, \qquad A_{\partial\mathcal{O},\mathcal{V}} = 1 + \frac{\Delta_\mathcal{V}(\Delta_\mathcal{V} - 3)}{6\Delta_\mathcal{O}} \,. \tag{10}$$

We now proceed to the numerical check. We add the correction $\mathcal{V} = \epsilon$ to the CFT Hamiltonian. We consider the same 4 energy gaps as in Section 2.1, corresponding to $\sigma, \partial\sigma, \epsilon, \partial\epsilon$. We test equations (2) for these states, replacing ... by the correction terms $\delta E_i$ given in (10). These corrections can all be evaluated in terms of the scaling dimensions $\Delta_\sigma, \Delta_\epsilon$ and the OPE coefficients $f_{\sigma\epsilon\sigma}, f_{\epsilon\epsilon\epsilon}$, which are all known, see App. B, times an effective coupling $g_\epsilon$ which we treat as a free parameter. We perform the fit by minimizing the following quantity over $\alpha$ and $g_\epsilon$:

$$\delta = \sum_i (\alpha \hat{E}_i - \Delta_i - \delta E_i)^2 \,. \tag{11}$$

The result of this exercise are shown in Fig. 5. Compared to Fig. 3, the fit works much better. The corrected energy levels are almost constant with $h$ in the shown window, while in Fig. 3 the levels had a linear "drift". This drift is now corrected away, for all 4 states, by

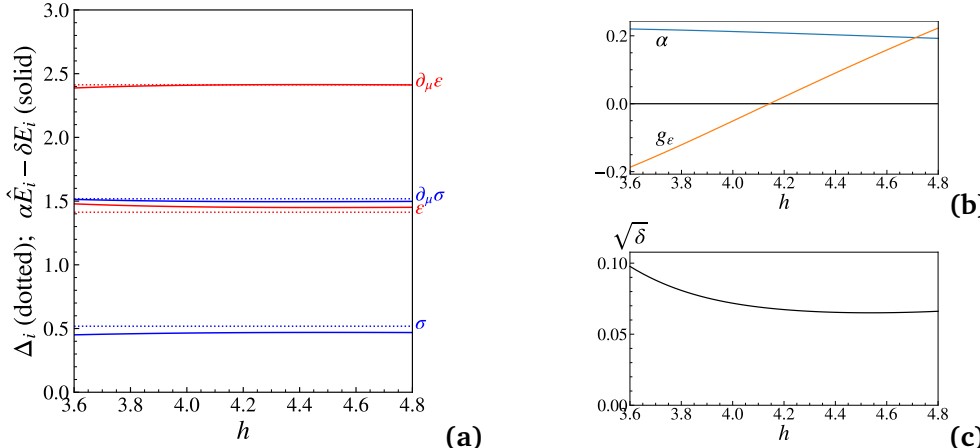

Figure 5: **(a)** Dotted: CFT states $\sigma$, $\partial_\mu\sigma$, $\epsilon$, $\partial_\mu\epsilon$. Solid: $\alpha\hat{E}_i - \delta E_i$ for the corresponding exact diagonalization energy gaps, with the best fit $\alpha, g_\epsilon$. **(b)** Best fit $\alpha, g_\epsilon$. **(c)** Fit quality.

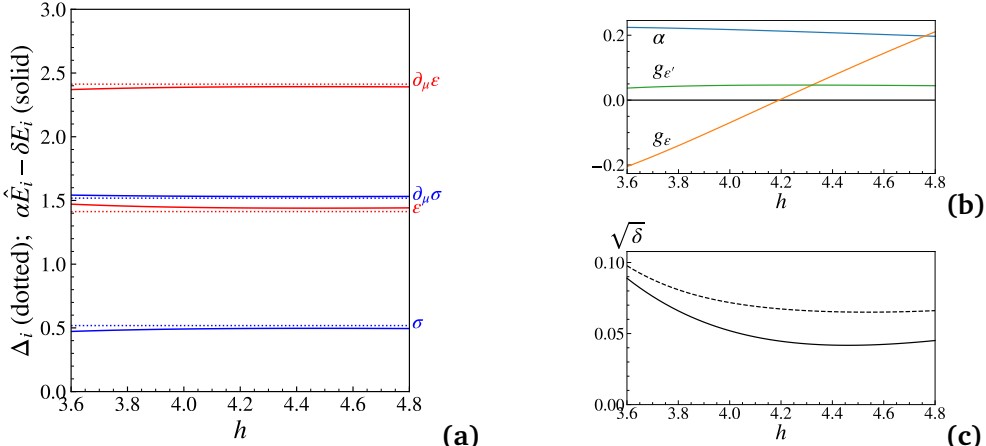

Figure 6: **(a)** Dotted: CFT states $\sigma, \partial_\mu\sigma, \epsilon, \partial_\mu\epsilon$. Solid: $\alpha\hat{E}_i - \delta E_i$ for the corresponding exact diagonalization energy gaps, with the best fit $\alpha, g_\epsilon, g_{\epsilon'}$. **(b)** Best fit $\alpha, g_\epsilon, g_{\epsilon'}$. **(c)** Solid: fit quality. Dashed: fit quality from Fig. 3, for comparison.

adding the effect of the coupling $g_\epsilon$, which itself depends almost linearly on $h$. Remarkably, $g_\epsilon$ crosses 0 around $h = h_c \approx 4.15$. It is positive at $h > h_c$ and negative at $h < h_c$. This is in full agreement with the expectation that the critical point should have $g_\epsilon = 0$, while positive/negative $g_\epsilon$ should correspond to the phase of preserved/broken $\mathbb{Z}_2$ invariance.

## 4.2 Adding $\epsilon'$ perturbation

To fix the discrepancy between the numerical data and the CFT spectrum in Fig. 5, let us try to add the next scalar perturbation, which is the leading irrelevant scalar $\epsilon'$, $\Delta_{\epsilon'} \approx 3.83$. While $g_\epsilon$ varies linearly and crosses zero at the critical point, the new effective coupling $g_{\epsilon'}$ is expected to be essentially constant in $h$ and therefore has a chance to fix the discrepancy in Fig. 5 which is nearly $h$ independent.

For this new test we consider the same 4 states as in Fig. 5. We introduce corrections $\delta E_i$ which are the sums of the corrections (10) for $\mathcal{V} = \epsilon, \epsilon'$ with independent couplings $g_\epsilon, g_{\epsilon'}$. The OPE coefficients $f_{\mathcal{O}\epsilon'\mathcal{O}}$ are known (App. B) and the new correction can be evaluated. We minimize (11) over $\alpha, g_\epsilon, g_{\epsilon'}$. The results are shown in Fig. 6.

We see in that figure that indeed the coupling $g_{\epsilon'}$ is almost constant. Although the fit quality is improved by about 30%, it remains imperfect. We will now see how to solve the remaining discrepancy by adding the spin-4 perturbation $C_{\mu\nu\lambda\sigma}$.

## 4.3 Adding $C_{\mu\nu\lambda\sigma}$ perturbation

In this section we will add yet another perturbation to the CFT Hamiltonian. The total number of parameters in the fit will become 4. To make the game interesting, we need to consider more than 4 energy levels $\sigma, \partial\sigma, \epsilon, \partial\epsilon$ we considered so far. In this section we will consider energy levels corresponding to all descendants of $\sigma, \epsilon$ up to and including level 2. In CFT, these energy levels are $\Delta_{\mathcal{O}} + k$ with $\mathcal{O} = \sigma, \epsilon$ and $k = 0, 1, 2$. However in the exact diagonalization we expect that the level 2 descendants $\partial\partial\mathcal{O}$ split as 1+5, corresponding to the dimensions of irreducible $O(3)$ representations, which also remain irreducible under $I_h$. Thus we have the total of 8 energy levels.

Which new perturbation should we consider next? One possibility is $\epsilon''$ but this operator has very high dimension 6.8956(43) [14] so we judge its importance unlikely.

What about adding the stress tensor $T_{\mu\nu}$? In an exactly rotationally invariant setting, this perturbation is unimportant because it just rescales the radius of the sphere or, equivalently, rescales the units of time. This effect is already taken care of in our scheme via the speed of light parameter $\alpha$. Although our model is not rotationally invariant but only $I_h$ invariant, perturbations of scalar descendants feel this difference only starting from $k = 3$. This is related to the fact that the first non-isotropic invariant tensor of the $I_h$ group has six indices. Since here we are dealing with $k \leq 2$, we do not have to consider $T_{\mu\nu}$ perturbations. See App. C.2 for a more detailed discussion.

We are thus led to consider the perturbation related to the spin-4 primary operator $C_{\mu\nu\lambda\sigma}$, of dimension 5.022665(28). In Fig. 7 we show the results of the test where we fit the 8 above-mentioned energy levels. The cost function is (11) where $i = 1, \ldots, 8$ and

- $\hat{E}_1, \ldots, \hat{E}_4$ are the energy gaps corresponding to $\sigma, \epsilon, \partial\sigma, \partial\epsilon$ used in the previous plots.

- $\hat{E}_5, \hat{E}_6$ are the energy gaps corresponding to $\partial\partial\sigma$. These are the $\mathbb{Z}_2$-odd levels having multiplicity 1 and 5 in Fig. 1, located above the $\mathbb{Z}_2$-odd multiplicity-3 level of $\partial\sigma$.

- $\hat{E}_7, \hat{E}_8$ are the energy gaps corresponding to $\partial\partial\epsilon$. These are the $\mathbb{Z}_2$-even levels having degeneracy 1 and 5. There is some discrete choice to be made in assigning these levels, as they have to be distinguished from the multiplicity-1 level of $\epsilon'$ and the multiplicity-5 level of $T_{\mu\nu}$ which are also $\mathbb{Z}_2$-even and have a closeby scaling dimension. We chose the only assignment which leads to a satisfactory fit.

The correction terms $\delta E_i$ are now the sums of three terms, with independent couplings $g_\epsilon, g_{\epsilon'}, g_C$. Corrections due to $C$ are given in App. C.3. Results of the fit with only 2 couplings $g_\epsilon, g_{\epsilon'}$ are also shown in Fig. 7 for comparison.

We see that the fit is outstandingly good with 3 couplings and $\alpha$, while it is much worse with 2 couplings and $\alpha$ (not surprisingly because of the conclusions in Section 4.2).

It is especially remarkable that with 3 couplings and $\alpha$ we are able to reconcile the second level descendants of $\sigma$ and $\epsilon$ with the CFT data. In Fig. 1, we see that the 1+5 components of the CFT states $\partial\partial\sigma$ and $\partial\partial\epsilon$ have a significant splitting in the exact diagonalization spectrum. Our correction terms are able to account for this splitting very well. The correction term is especially large for $\partial^2\sigma$, due to $\sigma$ being close to the unitarity bound, see the discussion in App. (C.1.2).

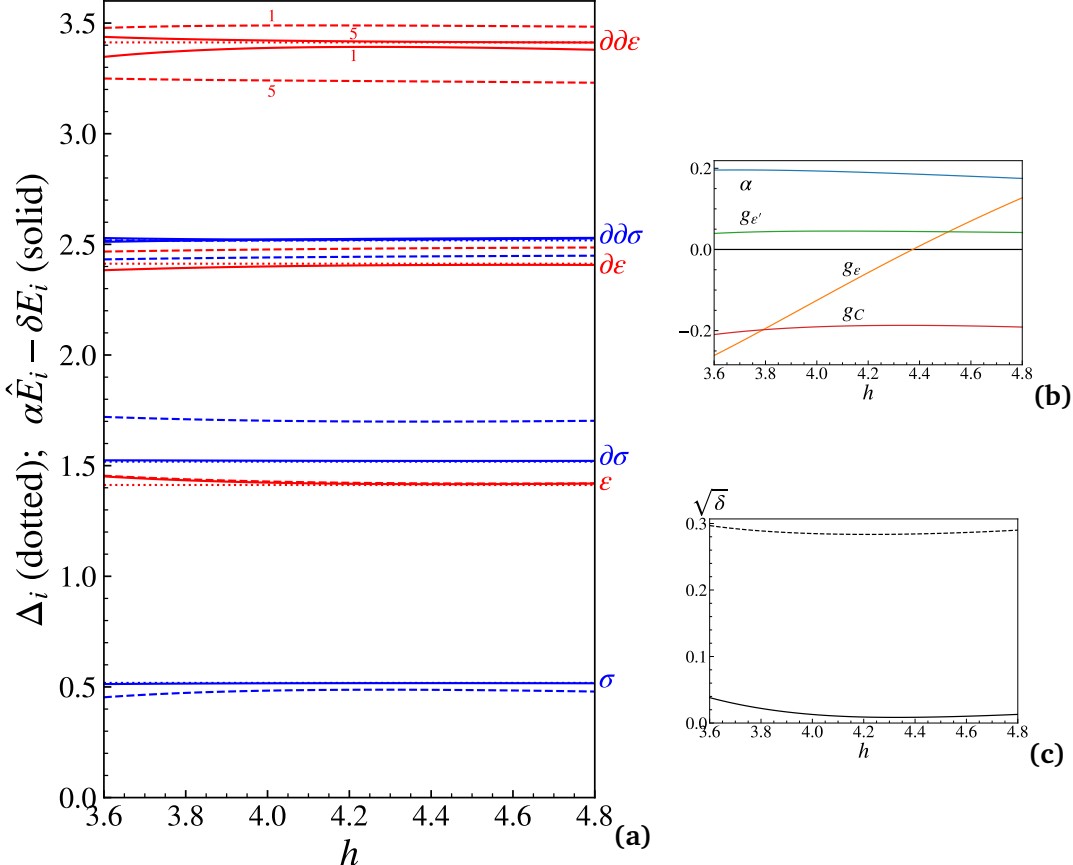

Figure 7: **(a)** Dotted: CFT states $\mathcal{O}, \partial\mathcal{O}, \partial\partial\mathcal{O}, \mathcal{O} = \sigma, \epsilon$. Solid: $\alpha\hat{E}_i - \delta E_i$ for the corresponding exact diagonalization energy gaps, with the best fit $\alpha, g_\epsilon, g_{\epsilon'}, g_C$. Dashed: the same but fitting only 3 parameters $\alpha, g_\epsilon, g_{\epsilon'}$. We show multiplicity (1 or 5) only for the $\partial\partial\epsilon$ levels. **(b)** Best fit $\alpha, g_\epsilon, g_{\epsilon'}, g_C$ for the 4-parameter fit. **(c)** Solid: 4-parameter fit quality. Dashed: 3-parameter fit quality.

## 4.4 Tests for $T_{\mu\nu}$ and $\epsilon'$ levels

Two more interesting energy levels to consider are the stress tensor $T \equiv T_{\mu\nu}$ and $\epsilon'$. These are $\mathbb{Z}_2$ even and have multiplicity 5 and 1, respectively. The corresponding exact diagonalization levels could be confused with the $\partial\partial\epsilon$ levels which are split as $1+5$, but we identified those in the previous section, so consequently we can now identify $T_{\mu\nu}$ and $\epsilon'$, see the labels in Fig. 1.

We would like to test the formula

$$\alpha\hat{E}_i = \Delta_i + \delta E_i, \tag{12}$$

for these two levels. For this test we take $\delta E_{\epsilon'}$ the sum of the $\epsilon$ and $\epsilon'$ corrections and $\delta E_T$ as the $\epsilon$ correction. We take $\alpha$ and the couplings $g_\epsilon, g_{\epsilon'}$ from Fig. 6(b).

The result of this test is shown in Fig. 8. The dashed lines show $\alpha\hat{E}_i$ with $\alpha$ from Fig. 6(b), i.e. just the speed of light rescaling. This agrees poorly with the CFT levels. The solid lines show $\alpha\hat{E}_i - \delta E_i$. This agrees much better with the CFT levels. The agreement is still imperfect, but the deviation is almost constant in $h$. This gives us hope that this deviation may be fixed by the corrections due to $C$ and, in case of $T$, due to $\epsilon'$. Indeed, coupling $g_C$ and $g_{\epsilon'}$ are almost constant in $h$ in the considered range, see Fig. 7(b). These corrections may not be evaluated at present since the corresponding OPE coefficients are not known.

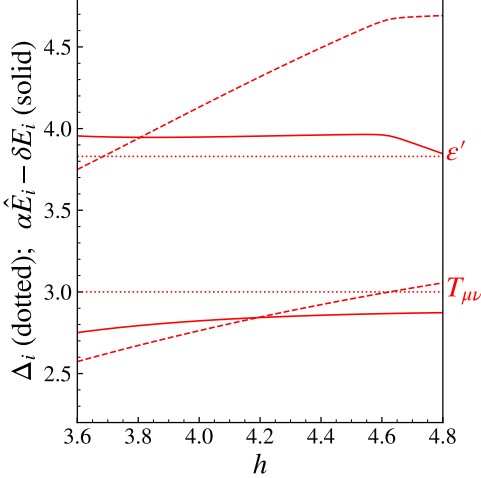

Figure 8: Dotted: CFT energy levels $\epsilon'$ and $T_{\mu\nu}$. Solid: $\alpha\hat{E}_i - \delta E_i$, with incomplete corrections (see the text). Dashed: $\alpha\hat{E}_i$. The "kink" in the $\epsilon'$ level curves at $h \sim 4.6$ is due to the avoided level crossing with another $\mathbb{Z}_2$ even singlet level.

## 4.5 Level-3 descendants

It was already almost a miracle that we managed to reproduce the level 2 descendants $\partial\partial\sigma$ and $\partial\partial\epsilon$, due to a significant splitting between the $1+5$ components. In this section, let us see what happens for level 3.

Level-3 descendants of a primary scalar $\mathcal{O}$ split as $\mathbf{3} + \mathbf{3'} + \mathbf{4}$, see App. C.1.3. The corresponding components for $\mathcal{O} = \sigma, \epsilon$ are identified in Fig. 1. We see that the splittings between the different components are large, especially for $\sigma$. Can we reproduce them?

As shown in App. C.1.3, the corrections for the component $\mathbf{3}$ of $\partial\partial\partial\mathcal{O}$ are sensitive to the same couplings $g_\epsilon, g_{\epsilon'}, g_C$ determined in Section 4.3 by fitting the descendants of level $k \le 2$. On the other hand the corrections for $\mathbf{3'}$ and $\mathbf{4}$ take the form

$$\delta E^{(4)}_{\partial\partial\partial\mathcal{O}} = \delta E^{(7)}_{\partial\partial\partial\mathcal{O}} + a\,, \qquad \delta E^{(3')}_{\partial\partial\partial\mathcal{O}} = \delta E^{(7)}_{\partial\partial\partial\mathcal{O}} - \frac{4}{3}a\,, \qquad (13)$$

where $\delta E^{(7)}$ depends on $g_\epsilon, g_{\epsilon'}, g_C$, while the splitting $a$ depends on extra couplings $\tilde{g}_\epsilon, \tilde{g}_{\epsilon'}, \tilde{g}_C$ which measure the deviation of the corresponding coupling functions $g(y)$ from constants, consistently with $I_h$ invariance.

Given that the splitting effect depends on three new couplings, it would be easy to fix those couplings to reproduce the splitting $\hat{E}^{(4)} - \hat{E}^{(3')}$, both for $\sigma$ and $\epsilon$. There is no great value in this observation, but we have checked that this could be done reasonably well even using just one coupling $\tilde{g}_\epsilon$.

It is more interesting to look at a new averaged variable

$$\hat{E}^{(7)} = \frac{4}{7}\hat{E}^{(4)} + \frac{3}{7}\hat{E}^{(3')}\,, \qquad (14)$$

which is not sensitive to the new couplings $\tilde{g}_\epsilon, \tilde{g}_{\epsilon'}, \tilde{g}_C$, at least to the first order as we are using here.

In Fig. 9 we do the following test. For $\mathcal{O} = \sigma, \epsilon$, we plot $\alpha\hat{E}_i - \delta E_i$ for the gaps corresponding to the component $\mathbf{3}$ of $\partial\partial\partial\mathcal{O}$ and for the linear combinations (14) of the $\mathbf{3'}$ and $\mathbf{4}$ components, which should not be sensitive to the splitting effect. The values of $\alpha, g_\epsilon, g_{\epsilon'}, g_C$ are taken from Fig. 7(b). We compare to the CFT levels $\partial\partial\partial\mathcal{O}$.

We see that the component $\mathbf{3}$ agrees rather well. On the other hand the linear combination $\hat{E}^{(7)}$ agrees well for $\epsilon$ but not for $\sigma$.

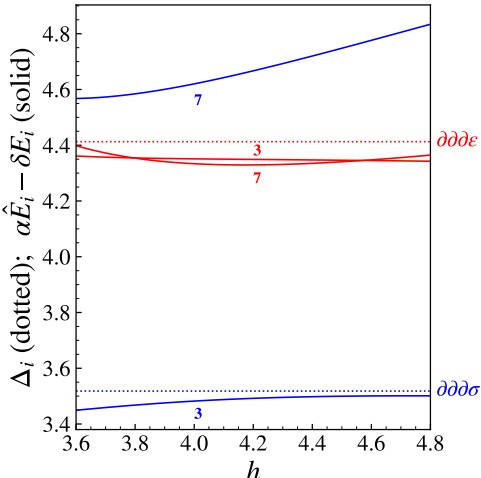

Figure 9: Dotted: CFT energy levels $\partial\partial\partial\sigma$, $\partial\partial\partial\epsilon$. Solid: $\alpha\hat{E}_i - \delta E_i$ for $\hat{E}_i$ corresponding to the component **3** of $\partial\partial\partial\mathcal{O}$ and for the linear combinations (14) (see the text).

We do not consider this disagreement as contradicting our framework. Indeed the splitting between **3'** and **4** for $\sigma$ is huge. Applying our first-order effective theory in this case is clearly stretching it beyond its range of validity.

## 5 Conclusions and outlook

At the first glance, the exact diagonalization spectrum of the icosahedron in Fig. 1 is a mess. In this paper we brought order to this chaos, relating this spectrum to the 3D Ising CFT spectrum by means of a first order effective theory perturbing the CFT Hamiltonian by integrals of local operators over the sphere. With just three effective couplings, and a speed of light parameter $\alpha$, we managed to reproduce very well 8 energy levels corresponding to the descendants of the CFT operators $\sigma$ and $\epsilon$ up to level 2, in a window of transverse magnetic fields around $h_c \sim 4.375$. We showed that the effective theory also tends to reproduce the energy levels of $\epsilon'$ and $T$, although the agreement is less than perfect, because not all corrections could be evaluated due to incomplete knowledge of CFT data.

We found that the effective theory stops being fully successful for level 3 descendants. This is not so surprising because the splittings between states which have to be exactly degenerate in CFT become huge at this level. As any effective field theory, ours has finite range of validity, and large splittings mark this range.

We thus consider Conjecture 1 demonstrated for the icosahedron model. For the fuzzy sphere model of [10], Conjecture 1 will be demonstrated in the upcoming paper [19].

In the icosahedron model, we have not attempted to extract new CFT data from exact diagonalization. The icosahedron model is too small and "dirty" for that. However, the game will become much more interesting for the fuzzy sphere model of [10]. When comparing to CFT, the authors of [10] used only the speed of light parameter $\alpha$. Indeed, finite $N$ corrections in that model were already very small. As discussed in the introduction, this may be attributed to a double tuning of model's parameters, which has the chance to tune to zero the coefficients of the relevant and of the leading irrelevant $\mathbb{Z}_2$ even scalar operators. When deviations are already small, applying effective theory should be an extremely lucrative enterprise. First of all, it will lead to further dramatic improvement in the agreement with CFT. Second, it will lead to new ways of determining the CFT data *from the deviations themselves*, since these deviations

are controlled, as we have seen, by universal formulas depending on the CFT parameters, up to a handful of couplings which must be determined from a fit. The noise will thus become the signal.

To give an idea, suppose that the deviations for CFT operators $\mathcal{O}_1, \mathcal{O}_2$ are proportional to the same effective coupling $g_{\mathcal{V}} \int_{S^2} \mathcal{V}$. Then, by measuring the deviation ratio we can determine the ratio of the OPE coefficients $f_{\mathcal{O}_1 \mathcal{V} \mathcal{O}_1} / f_{\mathcal{O}_2 \mathcal{V} \mathcal{O}_2}$. If one of these OPE coefficients is known (e.g. from the bootstrap), we thus determine the other.

For the icosahedron model, the first-order conformal perturbation theory was sufficient to make the point. To fully leverage the power of the effective field theory for the fuzzy sphere model might require going to the second order in perturbing effective coupling. This promises a lot of interesting interplay between conformal field theory and exact diagonalization spectroscopy in the near future.

As for the icosahedron model, it would be interesting to consider its generalization, adding to the TFIM Hamiltonian (1) an isotropic Heisenberg nearest neighbor coupling $J_1$. This does not change the universality class of the transition. Tuning $J_1$ one could then tune the effective conformal perturbation theory couplings, making them potentially smaller than in the TFIM model considered in this paper. This would then be analogous to what presumably happens in the fuzzy sphere model of [10].[8]

## Acknowledgments

SR thanks Benoit Sirois for collaboration at the early stages of this work. BL thanks Benoit Sirois, Ning Su, Zechuan Zheng and especially Junchen Rong for useful discussions. BL thanks the IHES for hospitality. SR thanks Jesper Jacobsen for references about finite size scaling in (1+1)D.

**Funding information**    This work is supported by the Simons Foundation grant 733758 (Simons Bootstrap Collaboration).

## A    Irreducible representations of the icosahedral group

In this appendix we introduce the irreducible representations of the proper icosahedral group $I \subset SO(3)$ which is isomorphic to $A_5$. The full icosahedral group $I_h = I \times \mathbb{Z}_2^{O(3)}$.

There are five irreducible representations, labeled by their dimensions **1** (trivial representation), **3**, **3'**, **4** and **5**. We can think of **3** and **5** as the vector and the symmetric traceless two-index tensor representations of $SO(3)$, which remain irreducible under $I$. On the other hand, the 7-dimensional symmetric traceless three-index tensor of $SO(3)$ splits as **3'** + **4** under $I$.

See Table 1 for the character table of $I$.

## B    3D Ising CFT

In this appendix we collect known 3D Ising CFT data used in this work. Primary operators of the 3D Ising CFT are characterized by their scaling dimension $\Delta$, spin $\ell$, and $\mathbb{Z}_2$ and $\mathbb{Z}_2^{O(3)}$ quantum numbers. All operators we need have $\mathbb{Z}_2^{O(3)} = 1$. Scaling dimensions of primary

---

[8]We thank Andreas Läuchli for this suggestion.

operators and their OPE coefficients are shown in Tables [2]. In addition, we have

$$f_{T\epsilon T} = 0.8658(69) \quad [11]. \tag{B.1}$$

This was also determined in [23] who found $f_{T\epsilon T} = 0.87(6)$ (in the normalization of [11]).

Stress tensor OPE coefficient $f_{\mathcal{O}T\mathcal{O}}$, for $\mathcal{O}$ a primary scalar and $T$ canonically normalized, is given by

$$f_{\mathcal{O}T\mathcal{O}} = -\frac{d\Delta_{\mathcal{O}}}{d-1}\frac{1}{\text{Vol}(S^2)}. \tag{B.2}$$

This is in the normalization where the 3pt function is given by $\left(Z^\mu = \frac{x_{13}^\mu}{x_{13}^2} - \frac{x_{23}^\mu}{x_{23}^2}\right)$

$$\langle \mathcal{O}(x_1)\mathcal{O}(x_2)T^{\mu\nu}(x_3)\rangle = \frac{f_{\mathcal{O}T\mathcal{O}}}{|x_{12}|^{2\Delta_{\mathcal{O}}-1}|x_{23}||x_{31}|}\left(Z^\mu Z^\nu - \frac{\delta^{\mu\nu}}{3}Z^\rho Z_\rho\right). \tag{B.3}$$

## C  Matrix elements

In this appendix we will explain how to evaluate the energy correction (8). In this paper we will need this formula for $\psi$ corresponding to $\mathcal{O}$ a scalar primary or its descendants up to level 3, as well as for $\psi$ the stress tensor. We will focus on these cases. The perturbation $\mathcal{V}$ will be a primary scalar or a symmetric traceless primary up to spin 4.

The basic idea to compute $\langle\psi|\delta H|\psi\rangle$ is that we can map $\mathbb{R}\times S^2$ to $\mathbb{R}^3$ via $r = e^\tau$. Then the $\tau$ direction becomes the radial direction in $\mathbb{R}^3$. The bra and ket states $\psi$ are mapped to operator insertions at $\infty$ and 0, while $\mathcal{V}$ has to be integrated over the unit sphere.

### C.1  Scalar $\mathcal{V}$ and $\mathcal{O}$

The most basic case is when $\mathcal{O}$ and $\mathcal{V}$ are scalar primaries. Then we have:

$$\langle \mathcal{O}|\delta H|\mathcal{O}\rangle = \int_{|y|=1} g(y)\langle\mathcal{O}(\infty)\mathcal{V}(y)\mathcal{O}(0)\rangle, \tag{C.1}$$

where we abuse the notation and use $y$ to parameterize the unit sphere embedded in $\mathbb{R}^3$. As usual we have $\mathcal{O}(\infty) := \lim_{w\to\infty}|w^2|^{\Delta_{\mathcal{O}}}\mathcal{O}(w)$. From the known expression for the CFT 3pt function $\langle\mathcal{O}(w)\mathcal{V}(y)\mathcal{O}(x)\rangle$, we have $\langle\mathcal{O}(\infty)\mathcal{V}(y)\mathcal{O}(0)\rangle = f_{\mathcal{O}\mathcal{V}\mathcal{O}}$, independently of $y$, $|y| = 1$. Integrating over the sphere and taking into account that the primary state is unit normalized we get, for $\mathcal{V},\mathcal{O}$ scalars:

$$\delta E_{\mathcal{O}} = g_{\mathcal{V}}f_{\mathcal{O}\mathcal{V}\mathcal{O}}, \qquad g_{\mathcal{V}} := \int_{S^2} g(y). \tag{C.2}$$

Table 1: Character table of $I \cong A_5$. There are five irreducible representation, denoted by their dimensions. $\varphi = \frac{\sqrt{5}+1}{2}$.

| $A_5$ conjugacy class | $[e]$ | $[(1,2)(3,4)]$ | $[(1,2,3)]$ | $[(1,2,3,4,5)]$ | $[(1,2,3,5,4)]$ |
|---|---|---|---|---|---|
| size | 1 | 15 | 20 | 12 | 12 |
| $SO(3)$ representative | $\begin{pmatrix} 1 & 0 & 0 \\ 0 & 1 & 0 \\ 0 & 0 & 1 \end{pmatrix}$ | $\begin{pmatrix} -\frac{1}{2} & \frac{1}{2\varphi} & \frac{\varphi}{2} \\ \frac{1}{2\varphi} & -\frac{\varphi}{2} & \frac{1}{2} \\ \frac{\varphi}{2} & \frac{1}{2} & \frac{1}{2\varphi} \end{pmatrix}$ | $\begin{pmatrix} -\frac{1}{2} & -\frac{1}{2\varphi} & \frac{\varphi}{2} \\ \frac{1}{2\varphi} & \frac{\varphi}{2} & \frac{1}{2} \\ -\frac{\varphi}{2} & \frac{1}{2} & -\frac{1}{2\varphi} \end{pmatrix}$ | $\begin{pmatrix} -\frac{1}{2\varphi} & \frac{\varphi}{2} & -\frac{1}{2} \\ \frac{\varphi}{2} & \frac{1}{2} & \frac{1}{2\varphi} \\ \frac{1}{2} & -\frac{1}{2\varphi} & -\frac{\varphi}{2} \end{pmatrix}$ | $\begin{pmatrix} \frac{\varphi}{2} & -\frac{1}{2} & -\frac{1}{2\varphi} \\ \frac{1}{2} & \frac{1}{2\varphi} & \frac{\varphi}{2} \\ -\frac{1}{2\varphi} & -\frac{\varphi}{2} & \frac{1}{2} \end{pmatrix}$ |
| $\chi_1$ | 1 | 1 | 1 | 1 | 1 |
| $\chi_3$ | 3 | $-1$ | 0 | $1-\varphi$ | $\varphi$ |
| $\chi_{3'}$ | 3 | $-1$ | 0 | $\varphi$ | $1-\varphi$ |
| $\chi_4$ | 4 | 0 | 1 | $-1$ | $-1$ |
| $\chi_5$ | 5 | 1 | $-1$ | 0 | 0 |

Table 2: $\mathbb{Z}_2, \ell, \Delta$ and OPE coefficients of a few low-lying primary operators of the 3D Ising CFT. Boldface errors are rigorous, other errors are nonrigorous from the extremal functional method as explained in [14]. Results from [24] have no error bar. U stands for unknown.

| $\mathcal{O}$ | $\mathbb{Z}_2$ | $\ell$ | $\Delta$ | $f_{\sigma\mathcal{O}\sigma}$ | $f_{\epsilon\mathcal{O}\epsilon}$ | $f_{\epsilon'\mathcal{O}\epsilon'}$ |
|---|---|---|---|---|---|---|
| $\sigma$ | $-$ | 0 | 0.5181489(**10**) [13] | 0 | 0 | 0 |
| $\epsilon$ | $+$ | 0 | 1.412625(**10**) [13] | 1.0518537(**41**) [13] | 1.532435(**19**) [13] | 2.3956 [24] |
| $\epsilon'$ | $+$ | 0 | 3.82968(23) [14] | 0.053012(55) [14] | 1.5360(16) [14] | 7.6771 [24] |
| | | | 3.82951(**61**) [16] | 0.05304(**16**) [16] | 1.5362(**12**) [16] | |
| $\epsilon''$ | $+$ | 0 | 6.8956(43) [14] | 0.0007338(31) [14] | 0.1279(17) [14] | U |
| $T$ | $+$ | 2 | 3 | | | |
| $T'$ | $+$ | 2 | 5.50915(44) [14] | 0.0105745(42) [14] | 0.69023(49) [14] | U |
| $C$ | $+$ | 4 | 5.022665(28) [14] | 0.069076(43) [14] | 0.24792(20) [14] | U |

### C.1.1 $\partial\mathcal{O}$

We next discuss the case of descendants of $\mathcal{O}$. For the first level descendants we have to evaluate:

$$\langle \partial_\mu\mathcal{O}| \,\delta H \,|\partial_\nu\mathcal{O}\rangle = \int_{|y|=1} g(y)\langle P_\mu\mathcal{O}|\mathcal{V}(y)|P_\nu\mathcal{O}\rangle \,, \tag{C.3}$$

where in the r.h.s. we have states in the radial quantization. The matrix element in the r.h.s. can be evaluated in two equivalent ways. The first way is to write

$$\langle P_\mu\mathcal{O}|\mathcal{V}(y)|P_\nu\mathcal{O}\rangle = \langle\mathcal{O}|K_\mu\mathcal{V}(y)P_\nu|\mathcal{O}\rangle \,, \tag{C.4}$$

and use commutation relations of the conformal algebra. The second way is to relate the computation to the known expression for the 3pt function, by writing:

$$\langle P_\mu\mathcal{O}|\mathcal{V}(y)|P_\nu\mathcal{O}\rangle = \lim_{x\to 0}\partial_{x^\mu}\langle\mathcal{O}^\dagger(x)\mathcal{V}(y)\partial_\nu\mathcal{O}(0)\rangle \,, \tag{C.5}$$

where $\mathcal{O}^\dagger(x) := |x^2|^{-\Delta_\mathcal{O}}\mathcal{O}(x^\mu/x^2)$ is the inversion applied to $\mathcal{O}$.

Let us discuss next the consequences of the icosahedral symmetry $I_h$. Since $\delta H$ is $I_h$ invariant, we must have:

$$\langle \partial_\mu\mathcal{O}| \,\delta H \,|\partial_\nu\mathcal{O}\rangle = B\delta_{\mu\nu} \,. \tag{C.6}$$

Indeed in general this matrix element has to be an invariant 2-tensor of $I_h$. Since $I_h$ is an irreducible subgroup of $O(3)$, all such tensors are multiples of $\delta_{\mu\nu}$. The constant $B$ can be found contracting (C.3) with $\delta_{\mu\nu}$. Then in the r.h.s. we will get $\langle P_\mu\mathcal{O}|\mathcal{V}(y)|P^\mu\mathcal{O}\rangle$, which is a constant on the sphere. Therefore, $B$ will be proportional to $g_\mathcal{V}$, i.e. insensitive to the deviations of $g(y)$ from its average value.

The final point is that we also have to compute the normalization of the descendant states, i.e. the constant $\mathcal{N}$ in

$$\langle P_\mu\mathcal{O}|P_\nu\mathcal{O}\rangle = \mathcal{N}\delta_{\mu\nu} \,. \tag{C.7}$$

This can be computed as in (C.4) or (C.5) setting $\mathcal{V} = 1$.

Putting all these ingredients together, we find, for $\mathcal{V}, \mathcal{O}$ scalars:

$$\delta E_{\partial\mathcal{O}} = g_\mathcal{V} f_{\mathcal{O}\mathcal{V}\mathcal{O}} A_{\partial\mathcal{O},\mathcal{V}} \,, \qquad A_{\partial\mathcal{O},\mathcal{V}} = 1 + \frac{C_\mathcal{V}}{6\Delta_\mathcal{O}} \,, \tag{C.8}$$

where $C_\mathcal{V} = \Delta_\mathcal{V}(\Delta_\mathcal{V} - 3)$ is the quadratic Casimir eigenvalue of a scalar primary.

Note that all three states $\partial\mathcal{O}$ get the same energy correction. This is related to the fact that the vector representation of $O(3)$ remains irreducible under $I_h$.

### C.1.2 $\partial\partial\mathcal{O}$

The computation for $\partial_\mu\partial_\nu\mathcal{O}$ is done similarly to $\partial_\mu\mathcal{O}$ but there is an interesting detail. In CFT all 6 states $\partial_\mu\partial_\nu\mathcal{O}$ are degenerate but they form two irreducible representations of $O(3)$ - the singlet $\partial^2\mathcal{O}$ and the traceless symmetric 2-tensor "$\partial_\mu\partial_\nu\mathcal{O} - \text{trace}$" with 5 states. These two representations remain irreducible under $I_h$. Thus we expect two different corrections for these two subspaces: $\delta E^{(1)}_{\partial\partial\mathcal{O}}$ and $\delta E^{(5)}_{\partial\partial\mathcal{O}}$.

The matrix element

$$\langle\partial_{\mu_1}\partial_{\mu_2}\mathcal{O}|\,\delta H\,|\partial_{\nu_1}\partial_{\nu_2}\mathcal{O}\rangle = \int_{|y|=1} g(y)\langle P_{\mu_1}P_{\mu_2}\mathcal{O}|\mathcal{V}(y)|P_{\nu_1}P_{\nu_2}\mathcal{O}\rangle\,, \tag{C.9}$$

is computed as for $\partial\mathcal{O}$ descendants. It should be an invariant tensor of $I_h$ with 4 indices and any such tensor is made of Kronecker $\delta$'s. This implies that, as for $\partial\mathcal{O}$, we can reduce the computation to rotationally invariant matrix elements and function $g(y)$ will enter only via $g_\mathcal{V}$. We omit the details and only give the final result, for $\mathcal{V},\mathcal{O}$ scalars:

$$\delta E^{(1)}_{\partial\partial\mathcal{O}} = g_\mathcal{V}f_{\mathcal{O}\mathcal{V}\mathcal{O}}A^{(1)}_{\partial\partial\mathcal{O},\mathcal{V}}\,, \qquad A^{(1)}_{\partial\partial\mathcal{O},\mathcal{V}} = 1 + \frac{C_\mathcal{V}^2 + C_\mathcal{V}(8\Delta_\mathcal{O}-2)}{12\Delta_\mathcal{O}(2\Delta_\mathcal{O}-1)}\,, \tag{C.10}$$

$$\delta E^{(5)}_{\partial\partial\mathcal{O}} = g_\mathcal{V}f_{\mathcal{O}\mathcal{V}\mathcal{O}}A^{(5)}_{\partial\partial\mathcal{O},\mathcal{V}}\,, \qquad A^{(5)}_{\partial\partial\mathcal{O},\mathcal{V}} = 1 + \frac{C_\mathcal{V}^2 + 10C_\mathcal{V}(2\Delta_\mathcal{O}+1)}{60\Delta_\mathcal{O}(1+\Delta_\mathcal{O})}\,. \tag{C.11}$$

It is worth pointing out that the $\delta E^{(1)}$ correction becomes singular in the limit $\Delta_\mathcal{O}\to 1/2$. This is because the norm of the singlet state goes to zero in this limit. This is related to the fact that the $\Delta_\mathcal{O} = 1/2$ scalar is free, hence $\partial^2\mathcal{O} = 0$. In the 3D Ising CFT, $\Delta_\sigma\approx 0.518$ is close to $1/2$. Hence, corrections for the singlet state $\partial^2\sigma$ are expected to be much larger than for the rest of level-2 descendants of $\sigma$.

### C.1.3 $\partial\partial\partial\mathcal{O}$

A new effect appears for the third level descendants $\partial_{\mu_1}\partial_{\mu_2}\partial_{\mu_3}\mathcal{O}$. There is a total of 10 CFT states in this level, split under $O(3)$ as $\mathbf{3} + \mathbf{7}$, which is the vector $\partial_\mu\partial^2\mathcal{O}$ plus the symmetric traceless 3-index tensor "$\partial_{\mu_1}\partial_{\mu_2}\partial_{\mu_3}\mathcal{O} - \text{traces}$". For a constant $g(y)$, $\delta H$ preserves $O(3)$ and we expect 2 independent corrections. For a non-constant $g(y)$, $\delta H$ only preserves $I_h$. Under $I_h$, $\mathbf{3}_{O(3)}$ remains irreducible and $\mathbf{7}_{O(3)}$ splits as $\mathbf{3}' + \mathbf{4}$. Thus for non-constant $g(y)$ we expect three independent corrections at this level—the first time we will see the difference between $O(3)$ and $I_h$ symmetry.

At the level of the matrix element

$$\langle\partial_{\mu_1}\partial_{\mu_2}\partial_{\mu_3}\mathcal{O}|\,\delta H\,|\partial_{\nu_1}\partial_{\nu_2}\partial_{\nu_3}\mathcal{O}\rangle = \int_{|y|=1} g(y)\langle P_{\mu_1}P_{\mu_2}P_{\mu_3}\mathcal{O}|\mathcal{V}(y)|P_{\nu_1}P_{\nu_2}P_{\nu_3}\mathcal{O}\rangle\,, \tag{C.12}$$

this is seen as follows. As usual this should be an invariant tensor of $I_h$. In addition to tensors built out of the Kronecker $\delta$, the group $I_h$ has one extra invariant six-index tensor $\mathfrak{S}$, see e.g. [25]. This tensor appears in the matrix element for non-constant $g(y)$, and this contribution causes the splitting between $\mathbf{3}'$ and $\mathbf{4}$ representations.

Equivalently, one may ask for which minimal spin $\ell$ the spin $\ell$ representation of $O(3)$ contains an $I_h$ singlet. This happens first for $\ell = 6$. Consider a function on the unit sphere given in terms of the above tensor $\mathfrak{S}$ (which is symmetric and can be chosen traceless) as

$$p_\mathfrak{S}(y) = \mathfrak{S}_{\mu_1\dots\mu_6}y^{\mu_1}\cdots y^{\mu_6}\,. \tag{C.13}$$

This function is $I_h$ invariant by construction but it belongs to the spin-6 representation of $O(3)$. In addition to $g_\mathcal{V}$ we define the second average coupling by

$$\tilde{g}_\mathcal{V} = \int_{S^2} g(y) p_\mathfrak{S}(y). \tag{C.14}$$

It is this coupling which will govern the splitting between $\mathbf{3}'$ and $\mathbf{4}$.

Omitting the details, the 3 corrections are given by, for $\mathcal{V}$ and $\mathcal{O}$ primary scalars:

$$\delta E^{(3)}_{\partial\partial\partial\mathcal{O}} = g_\mathcal{V} f_{\mathcal{O}\mathcal{V}\mathcal{O}} A^{(3)}_{\partial\partial\partial\mathcal{O},\mathcal{V}}, \tag{C.15}$$

$$\delta E^{(3')}_{\partial\partial\partial\mathcal{O}} = \delta E^{(7)}_{\partial\partial\partial\mathcal{O}} - \frac{4}{3} \tilde{g}_\mathcal{V} f_{\mathcal{O}\mathcal{V}\mathcal{O}} B_{\mathcal{O},\mathcal{V}}, \tag{C.16}$$

$$\delta E^{(4)}_{\partial\partial\partial\mathcal{O}} = \delta E^{(7)}_{\partial\partial\partial\mathcal{O}} + \tilde{g}_\mathcal{V} f_{\mathcal{O}\mathcal{V}\mathcal{O}} B_{\mathcal{O},\mathcal{V}}, \tag{C.17}$$

where

$$\delta E^{(7)}_{\partial\partial\partial\mathcal{O}} = g_\mathcal{V} f_{\mathcal{O}\mathcal{V}\mathcal{O}} A^{(7)}_{\partial\partial\partial\mathcal{O},\mathcal{V}}, \tag{C.18}$$

and

$$A^{(3)}_{\partial\partial\partial\mathcal{O},\mathcal{V}} = 1 + \frac{C_\mathcal{V}^3 + C_\mathcal{V}^2(22\Delta_\mathcal{O}+6) + 20C_\mathcal{V}(6\Delta_\mathcal{O}^2+2\Delta_\mathcal{O}-1)}{120\Delta_\mathcal{O}(\Delta_\mathcal{O}+1)(2\Delta_\mathcal{O}-1)}, \tag{C.19}$$

$$A^{(7)}_{\partial\partial\partial\mathcal{O},\mathcal{V}} = 1 + \frac{C_\mathcal{V}^3 + C_\mathcal{V}^2(42\Delta_\mathcal{O}+46) + 140C_\mathcal{V}(3\Delta_\mathcal{O}^2+6\Delta_\mathcal{O}+2)}{840\Delta_\mathcal{O}(\Delta_\mathcal{O}+1)(\Delta_\mathcal{O}+2)}, \tag{C.20}$$

$$B_{\mathcal{O},\mathcal{V}} = \frac{\Delta_\mathcal{V}^2(\Delta_\mathcal{V}+2)^2(\Delta_\mathcal{V}+4)^2}{\Delta_\mathcal{O}(\Delta_\mathcal{O}+1)(\Delta_\mathcal{O}+2)}. \tag{C.21}$$

The singularity of $A^{(3)}_{\partial\partial\partial\mathcal{O},\mathcal{V}}$ as $\Delta_\mathcal{O} \to 1/2$ has the same origin as for $A^{(1)}_{\partial\partial\mathcal{O},\mathcal{V}}$ in Eq. (C.10).

## C.2 Corrections from $\mathcal{V} = T_{\mu\nu}$

Let us discuss corrections when $\mathcal{V}$ is the stress tensor. We need to consider the matrix element

$$\int_{|y|=1} g_{\mu\nu}(y) \langle \psi'_k | T_{\mu\nu}(y) | \psi_k \rangle, \tag{C.22}$$

where $\psi_k$, $\psi'_k$ are any two states in the $\mathcal{O}$ multiplet at level $k \geq 0$, and $g_{\mu\nu}(y)$ is an $I_h$ invariant tensor function on the sphere. As discussed in Section 3, we may assume that $y^\mu g_{\mu\nu}(y) = 0$. We have

$$g_{\mu\nu}(y) = b(y_\mu y_\nu - \delta_{\mu\nu}) + (\mathfrak{S}) + \ldots, \tag{C.23}$$

where $b$ is a constant, $(\mathfrak{S})$ stands for $I_h$ invariant functions constructed by contracting $\mathfrak{S}_{\mu_1\ldots\mu_6}$ with powers of $y$, i.e. linear combinations of terms like

$$\mathfrak{S}_{\mu\nu\mu_1\ldots\mu_4} y_{\mu_1} \cdots y_{\mu_4}, \quad y_{(\mu}\mathfrak{S}_{\nu)\mu_1\ldots\mu_5} y_{\mu_1} \cdots y_{\mu_5}, \quad y_\mu y_\nu p_\mathfrak{S}(y), \quad \delta_{\mu\nu} p_\mathfrak{S}(y), \tag{C.24}$$

with $p_\mathfrak{S}(y)$ from (C.13), while $\ldots$ in (C.23) stands for terms involving higher invariant tensors of $I_h$.

The first term in (C.23) will give

$$b \int_{|y|=1} \langle \psi'_k | y^\mu y^\nu T_{\mu\nu}(y) | \psi_k \rangle. \tag{C.25}$$

The integral here is nothing but the CFT Hamiltonian (i.e. dilatation operator), giving the scaling dimension of the state. Thus the matrix element equals

$$b(\Delta_{\mathcal{O}} + k)\langle \psi'_k | \psi_k \rangle, \tag{C.26}$$

Since this correction is exactly proportional to the CFT energy of the state, it is equivalent to a small correction in the speed of light parameter $\alpha$ in our fits. Therefore, we do not have to consider it.

As for the terms $(\mathfrak{S}) + \dots$ in (C.23), they will not contribute to the matrix element (C.22) for $k \leq 2$, because the total number of indices in the states $\psi_k$, $\psi'_k$ is not enough to saturate the indices of $\mathfrak{S}$. For $k = 3$ the terms $(\mathfrak{S})$ will contribute and they will give rise to additional shifts to $3'$ and $4$, as in Section C.1.3. Omitting the details the result is given by:

$$\delta E^{(4)}_{\partial\partial\partial\mathcal{O}} = \tilde{g}_T \frac{f_{\mathcal{O}T\mathcal{O}}}{\Delta_{\mathcal{O}}(\Delta_{\mathcal{O}} + 1)(\Delta_{\mathcal{O}} + 2)}, \qquad \delta E^{(3')}_{\partial\partial\partial\mathcal{O}} = -\frac{4}{3}\delta E^{(4)}_{\partial\partial\partial\mathcal{O}}, \tag{C.27}$$

where $\tilde{g}_T$ is given by an integral of $g_{\mu\nu}(y)$ against a tensor function constructed out of $\mathfrak{S}$ (we won't need the exact expression). Recall that $f_{\mathcal{O}T\mathcal{O}} \propto \Delta_{\mathcal{O}}$, see App. B.

The relative size $(-4/3, 1)$ of these corrections is the same as in Section C.1.3, which is not accidental and can be interpreted via the Wigner-Eckart theorem.

## C.3 Corrections from $\mathcal{V} = C_{\mu\nu\lambda\sigma}$

The discussion has a lot of similarity to $\mathcal{V} = T_{\mu\nu}$. We consider the matrix element

$$\int_{|y|=1} g_{\mu\nu\lambda\sigma}(y)\langle \psi'_k | C_{\mu\nu\lambda\sigma}(y) | \psi_k \rangle, \tag{C.28}$$

where $\psi_k$, $\psi'_k$ are any two states in the $\mathcal{O}$ multiplet at level $k \geq 0$, and $g_{\mu\nu\lambda\sigma}(y)$ is an $I_h$ invariant tensor function on the sphere. As discussed in Section 3, we may assume that $y^\mu g_{\mu\nu\lambda\sigma}(y) = 0$. We have

$$g_{\mu\nu\lambda\sigma}(y) = \frac{g_C}{4\pi}(y_\mu y_\nu y_\lambda y_\sigma - \text{terms involving at least one } \delta) + (\mathfrak{S}) + \dots, \tag{C.29}$$

where $g_C$ is a constant and $(\mathfrak{S})$ stands for $I_h$ invariant functions constructed by contracting $\mathfrak{S}_{\mu_1\dots\mu_6}$ with powers of $y$ and $\dots$ with terms involving higher invariant tensors of $I_h$.

The first term in (C.23) contributes to the matrix element as

$$\frac{g_C}{4\pi}\int_{|y|=1}\langle \psi'_k | y^\mu y^\nu y^\lambda y^\sigma C_{\nu\nu\lambda\sigma}(y) | \psi_k \rangle. \tag{C.30}$$

Here the discussion deviates from $\mathcal{V} = T_{\mu\nu}$ where such a correction was equivalent to renormalizing $\alpha$, while here it is not, so we have to evaluate it carefully. This results in energy shifts, for levels $k \leq 3$:

$$\delta E_{\mathcal{O}} = g_C f_{\mathcal{O}C\mathcal{O}}, \qquad\qquad \delta E_{\partial\mathcal{O}} = g_C f_{\mathcal{O}C\mathcal{O}} A_{\partial\mathcal{O},C}, \tag{C.31}$$

$$\delta E^{(1)}_{\partial\partial\mathcal{O}} = g_C f_{\mathcal{O}C\mathcal{O}} A^{(1)}_{\partial\partial\mathcal{O},C}, \qquad\qquad \delta E^{(5)}_{\partial\partial\mathcal{O}} = g_C f_{\mathcal{O}C\mathcal{O}} A^{(5)}_{\partial\partial\mathcal{O},C}, \tag{C.32}$$

$$\delta E^{(3)}_{\partial\partial\partial\mathcal{O}} = g_C f_{\mathcal{O}C\mathcal{O}} A^{(3)}_{\partial\partial\partial\mathcal{O},C}, \qquad\qquad \delta E^{(7)}_{\partial\partial\partial\mathcal{O}} = g_C f_{\mathcal{O}C\mathcal{O}} A^{(7)}_{\partial\partial\partial\mathcal{O},C}. \tag{C.33}$$

Note that, as in the previous section, the energy shifts cause the split $6 \to 1 + 5$ for $k = 2$ and $10 \to 3 + 7$ for $k = 3$. We will give the expressions for the $A$ coefficients in the general case when $\mathcal{V}$ has spin $\ell$ (while $\ell = 4$ in the case at hand). The Casimir eigenvalue

$C_\mathcal{V} = \Delta_\mathcal{V}(\Delta_\mathcal{V} - 3) + \ell(\ell + 1)$. These coefficients generalize the previously given $\ell = 0$ expressions and they are given by:[9]

$$A_{\partial\mathcal{O},\mathcal{V}} = 1 + \frac{C_\mathcal{V}}{6\Delta_\mathcal{O}}, \tag{C.34}$$

$$A^{(1)}_{\partial\partial\mathcal{O},\mathcal{V}} = 1 + \frac{C_\mathcal{V}^2 + C_\mathcal{V}(8\Delta_\mathcal{O} - 2) - 4\ell(\ell + 1)(C_\mathcal{V} - (\ell + 2)(\ell - 1))}{12\Delta_\mathcal{O}(2\Delta_\mathcal{O} - 1)}, \tag{C.35}$$

$$A^{(5)}_{\partial\partial\mathcal{O},\mathcal{V}} = 1 + \frac{C_\mathcal{V}^2 + 10C_\mathcal{V}(2\Delta_\mathcal{O} + 1) + 2\ell(\ell + 1)(C_\mathcal{V} - (\ell + 2)(\ell - 1))}{60\Delta_\mathcal{O}(1 + \Delta_\mathcal{O})}, \tag{C.36}$$

$$A^{(3)}_{\partial\partial\partial\mathcal{O},\mathcal{V}} = 1 + \frac{C_\mathcal{V}^3 + C_\mathcal{V}^2(22\Delta_\mathcal{O} + 6) + 20C_\mathcal{V}(6\Delta_\mathcal{O}^2 + 2\Delta_\mathcal{O} - 1)}{120\Delta_\mathcal{O}(\Delta_\mathcal{O} + 1)(2\Delta_\mathcal{O} - 1)}$$
$$- \frac{4\ell(\ell + 1)(C_\mathcal{V} - (\ell + 2)(\ell - 1))(C_\mathcal{V} + 14\Delta_\mathcal{O} + 8)}{120\Delta_\mathcal{O}(\Delta_\mathcal{O} + 1)(2\Delta_\mathcal{O} - 1)}, \tag{C.37}$$

$$A^{(7)}_{\partial\partial\partial\mathcal{O},\mathcal{V}} = 1 + \frac{C_\mathcal{V}^3 + C_\mathcal{V}^2(42\Delta_\mathcal{O} + 46) + 140C_\mathcal{V}(3\Delta_\mathcal{O}^2 + 6\Delta_\mathcal{O} + 2)}{840\Delta_\mathcal{O}(\Delta_\mathcal{O} + 1)(\Delta_\mathcal{O} + 2)}$$
$$+ \frac{2\ell(\ell + 1)(C_\mathcal{V} - (\ell + 2)(\ell - 1))(3C_\mathcal{V} + 42\Delta_\mathcal{O} + 44)}{840\Delta_\mathcal{O}(\Delta_\mathcal{O} + 1)(\Delta_\mathcal{O} + 2)}. \tag{C.38}$$

For $k \leq 2$ the total number of indices in the states $\psi_k$, $\psi'_k$ is not enough to saturate the indices of $\mathfrak{S}$. Therefore the terms $(\mathfrak{S}) + \ldots$ in (C.23) will not contribute to the matrix element (C.22) for $k \leq 2$. For $k = 3$ the terms $(\mathfrak{S})$ will contribute and they will give rise to additional shifts to $3'$ and $4$ given by

$$\delta E^{(4)}_{\partial\partial\partial\mathcal{O}} = \tilde{g}_C \frac{f_{\mathcal{O}C\mathcal{O}}}{\Delta_\mathcal{O}(\Delta_\mathcal{O} + 1)(\Delta_\mathcal{O} + 2)}, \qquad \delta E^{(3')}_{\partial\partial\partial\mathcal{O}} = -\frac{4}{3}\delta E^{(4)}_{\partial\partial\partial\mathcal{O}}, \tag{C.39}$$

where the coupling $\tilde{g}_C$ is given by an integral of $g_{\mu\nu\lambda\sigma}(y)$ involving $\mathfrak{S}$, and we also include into $\tilde{g}_C$ some $\Delta_C$ dependent factors (we won't need the exact expression).

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
