# Peer review of "D Ising CFT and Exact Diagonalization on Icosahedron: The Power of Conformal Perturbation Theory"

_SciPost Physics, doi:SciPost Phys. 15, 243 (2023)_

## Round 2 · Referee Report · Anonymous (Referee 1) · 2023-10-14

Strengths

Please see report.

Weaknesses

Please see report.

Report

The main point of this nice paper is that finite-size effects in regulated CFT on a sphere can be understood using conformal perturbation theory.

It was clearly inspired by the recent development by Yin-Chen He and collaborators of a new rotation-invariant regulator for 2+1d CFT on $S^2$ in the form of the fuzzy sphere. The results so obtained agree amazingly well with conformal bootstrap results (and even extend some of them) even for very small systems. This paper gives an answer to the important question of why it works so well. The answer is that it is because of the spectrum of operators of the 2+1d Ising CFT, namely the finite-size effects are suppressed by the fact that the leading allowed irrelevant operator has a rather high dimension. Therefore, such nice results should hold for a larger class of regulators. The authors illustrate this point using a simple real-space regulator in the form of the transverse-field Ising model on an icosahedron, for whose spectrum they give a very convincing account.

As the authors point out, a good understanding of the finite-size effects will make the fuzzy-sphere technique all the more powerful.

It is quite wonderful (if unsurprising in retrospect) that the analysis depends not only on the spectrum of dimensions but also on the OPE coefficients.

I like it!

It is possible that the title is too humble, in the sense that the role of the icosahedron model is as a test case of a much more general method of analysis applicable to any regularized CFT. This may lead some potential readers to miss the nice point.

-- p.2 "results were obtain"
-- p.2 "The icosahedron is chosen because its spatial symmetry group is the largest irreducible subgroup of O(3)." I guess it should be the largest irreducible discrete subgroup, in that SO(2) is larger?
-- A frivolous comment: In the nice Figure 1, I would greatly prefer that the $Z_2$ even states were blue and the odd ones were red!

-- It would be friendly to give a little more explanation of why descendants associated to operators made by taking derivatives of a primary in the $\tau$ direction can be ignored in the analysis of section 3.

-- p.10 missing close parens "(not surprisingly...".

-- At the end of section 4.4 the authors say that certain corrections cannot be evaluated yet because certain OPE coefficients are not yet known. With enough confidence in this method, I suppose one could use it to estimate them. I see now that this is what the authors suggest in the conclusions for the fuzzy sphere case.

Requested changes

Please see report.

---

## Round 2 · Referee Report · Anonymous (Referee 2) · 2023-11-15

Report

In this paper, the authors study the $2+1$-dimensional transverse field Ising model with spins located on the vertices of a regular icosahedron. The authors then argue that the resulting energy spectrum, in the vicinity of the critical point described by the 3d Ising CFT, can be understood in terms of conformal perturbation theory, with deformations of the CFT given by integrals of local operators over the two-dimensional sphere. To test this conjecture, they compute the spectrum as a function of the transverse magnetic field via exact diagonalization and fit 8 of the lowest energy eigenvalues (above the ground state) with a 4-parameter effective theory: the Ising CFT (with a single "speed of light" parameter setting the overall energy normalization) plus the 3 lowest $\mathbb{Z}_2$-even deformations (computed to first order in perturbation theory).

The results are self-consistent, with the extracted coefficient of the one relevant deformation $\varepsilon$ varying approximately linearly with the magnetic field, while the coefficients of the other irrelevant deformations are approximately constant. As a further check, the authors show that the same fit parameters match the numerical results for two additional eigenvalues (corresponding to the primary operators $T_{\mu\nu}$ and $\varepsilon'$). This effective description appears to break down when applied to even higher eigenvalues (corresponding to level-3 descendants of $\sigma$), where higher-order effects are clearly necessary.

The paper is clear, thorough, and demonstrates the impressive effectiveness of conformal perturbation theory, even for such a coarse-grained lattice containing few spins. The main motivation for this paper is the recent remarkably precise 3d Ising CFT data obtained via "fuzzy sphere'' regularization, which the authors argue can also be understood in terms of conformal perturbation theory (though this is left for future work). This work therefore provides a useful lens with which to understand this newly developing line of research.

Requested changes

My comments are relatively minor:

1) I would like some quick clarification on what is meant by "Weyl-equivalent'' on page 1? If they mean that correlation functions on $\mathbb{R}^d$ transform covariantly when mapped to $\mathbb{R} \times S^{d-1}$, it is my understanding that this has only been proven (in https://arxiv.org/abs/1702.07079) for unitary CFTs in $d \leq 10$. Of course this would include the 3d Ising CFT, but I think it would be useful to clarify this statement.

2) For the case of an icosahedral lattice considered in this work, it's clear why the vicinity of the critical point should be described solely in terms of local deformations of the Ising CFT. However, it is much less clear to me why the effective description of the fuzzy sphere regularization considered in [9] can be described solely in terms of local operators. I realize the authors plan to consider this example in future work, but given that the main conjecture of this work is focused on the results of [9], could the authors provide a bit more explanation for their intuition that the fuzzy sphere regularization should be understood as a local cutoff?

3) Finally, there is a small typo in the paragraph below eq.~(4.3). I believe it should say "... and the OPE coefficients $f_{\sigma\varepsilon\sigma}$, $f_{\varepsilon\varepsilon\varepsilon}$, which are all known ...''.

Once these comments have been addressed, I will gladly recommend this paper for publication in SciPost Physics.

---

## Round 3 · Author Response

We thank both referees for their reports. We corrected the typos they noticed, thanks, and tried to address the comments (except for inverting the blue and red colors in Fig.1 - but we do agree that in other contexts predominance of blue would be greatly desirable).
The new footnote 7 hopefully addresses the request of Report 1 about derivatives of a primary in the tau direction. We also followed the advice of Report 1 and augmented the title to better reflect the scope of our work.
The new footnotes 1 and 4 hopefully address comments 1) and 2) of Report 2.
The new footnote 7 hopefully addresses the request of Report 1 about derivatives of a primary in the tau direction. We also followed the advice of Report 1 and augmented the title to better reflect the scope of our work.
The new footnotes 1 and 4 hopefully address comments 1) and 2) of Report 2.

---

## Editorial Decision

published